# Causally Disentangled World Models: Guiding Exploration with an Agency Bonus

## Abstract

Model-Based Reinforcement Learning (MBRL) promises to improve sample efficiency, yet conventional world models learn a purely observational, black-box model of dynamics. This leads to causal confounding—a failure to distinguish the environment's autonomous evolution from agent-induced interventions—resulting in poor generalization and inefficient exploration. To resolve this, we introduce the **Causal Disentanglement World Model (CDWM)**, which learns the necessary *interventional* model by imposing a structural causal assumption. Our dual-path architecture decomposes state transitions into an uncontrollable **Environment Pathway** and a controllable **Intervention Pathway**, making causal effects identifiable from observational data. Building on this, we derive the **Agency Bonus**, a principled intrinsic reward that quantifies the agent's causal influence to guide exploration. Extensive experiments on the Atari100k benchmark show CDWM achieves state-of-the-art performance, outperforming prior methods in sample efficiency and planning accuracy. Ablation studies confirm the architecture's adaptability and the exploration mechanism's effectiveness in sparse-reward settings. Our results establish that imposing a causal structure is a critical step toward building more robust, interpretable, and generalizable world models.

## 1 Introduction

Model-Based Reinforcement Learning (MBRL) (Luo et al., 2022) seeks to learn a **world model**—an internal simulator of the environment's dynamics—to address the notorious sample inefficiency of predominant model-free methods like DQN (Mnih et al., 2013; Fan et al., 2020) and PPO (Schulman et al., 2017). By allowing an agent to plan future actions by "imagining" their consequences, world models can drastically improve data efficiency (Ha & Schmidhuber, 2018a) and have become a driving force in intelligent decision-making (Zhao et al., 2019; Li et al., 2016; Kober et al., 2013; Mnih et al., 2015; François-Lavet et al., 2018; He et al., 2015; Mnih et al., 2013). This paradigm, exemplified by state-of-the-art methods such as Dreamer (Hafner et al., 2020) and MuZero (Schrittwieser et al., 2020; Sutton, 1990), has achieved remarkable performance. Yet, a fundamental limitation undermines their potential for true generalization. Existing world models typically learn a monolithic, black-box function $g(s_t, a_t) \rightarrow s_{t+1}$ optimized to predict the next state. While effective at statistical pattern matching, this approach learns a purely **correlational** model, failing to capture the underlying **causal** mechanisms that govern the world.

This failure to distinguish correlation from causation leads to a critical flaw: **causal confounding**. The model cannot disentangle changes endogenous to the environment from those exogenously forced by the agent's actions. As we formalize in Section 2, this creates a fundamental discrepancy between the **observational distribution** the model learns and the **interventional distribution** a planner truly needs (Pearl, 2009). This confounding manifests in two primary failures: (1) **Poor generalization under policy shifts**, where learned correlational patterns break and destabilize planning algorithms like MCTS (Kocsis & Szepesvári, 2006). (2) **Inefficient exploration**, as the agent cannot perform causal credit assignment to identify actions with meaningful impact, leading to wasteful, undirected exploration (Anonymous, 2021).

To overcome these challenges, we pivot from a purely statistical to a structural approach, reformulating the problem of world modeling through the lens of causal inference. Our central hypothesis is that world dynamics can be decomposed into two independent causal mechanisms: **autonomous**

**environmental dynamics** and **agent-driven interventions**. Based on this principle, we design and implement the **Causal Disentanglement World Model (CDWM)**. As illustrated in Figure 1, CDWM employs a novel dual-path architecture where an **Environment Pathway** learns to predict the action-independent evolution of the world, while an **Intervention Pathway** exclusively models the changes directly attributable to the agent's actions. This architectural enforcement of our causal assumption enables the identification of causal factors from purely observational data.

Furthermore, this explicit separation of causal influence uniquely enables a principled intrinsic motivation mechanism. We introduce the **Agency Bonus**, a cognitively inspired and causally informed intrinsic reward derived from the magnitude of the Intervention Pathway's output. This bonus encourages the agent to explore actions that have a tangible, measurable impact on the environment, systematically probing its own causal capabilities and dramatically improving exploration efficiency in sparse-reward settings.

We validate CDWM on the Atari100k benchmark (Ye et al., 2021a), where extensive experiments show it achieves state-of-the-art performance, outperforming prior methods in sample efficiency and planning accuracy. Our results establish that imposing a causal structure is a critical step toward building more robust, interpretable, and generalizable world models.

## 2 PROBLEM FORMULATION

The ultimate ambition of a Model-Based Reinforcement Learning (MBRL) agent is to learn a predictive world model that enables effective planning under any potential policy (Ha & Schmidhuber, 2018b). This requires the model to generalize beyond the specific experiences it was trained on, a challenge we formalize within the **Partially Observable Markov Decision Process (POMDP)** framework (Åström, 1965).

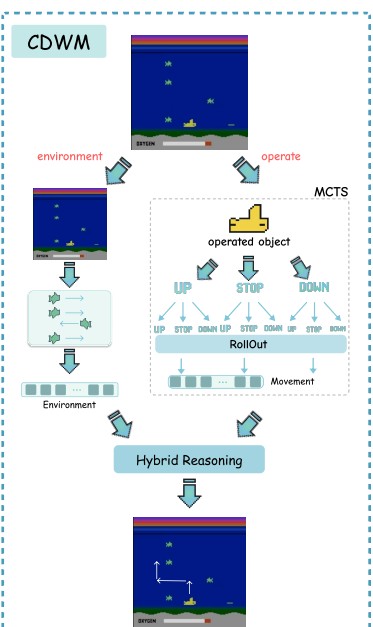

In this setting, an agent learns a latent dynamics model, $g_\phi$, by optimizing a predictive objective on a buffer of collected experience, $\mathcal{D}$. Conventionally, this is formulated as a maximum likelihood objective over the observed one-step transitions:

$$\min_\phi \quad \mathbb{E}_{(s_t, a_t, s_{t+1}) \sim \mathcal{D}}[-\log p_\phi(s_{t+1}|s_t, a_t)] \quad (1)$$

While this objective trains the model to be a proficient statistician of its past, it fails to equip it with the crucial attribute of a scientist: causal reasoning. A model trained via Eq. equation 1 alone (see Appendix F for a rigorous proof of identifiability under our assumptions). learns the purely observational conditional distribution $P(s_{t+1}|s_t, a_t)$, which is a tapestry of both true causal links and spurious, non-causal correlations.

To plan effectively, an agent does not need to know what state is likely to follow a given action in the past data. Instead, it must know what would happen if it were to force a particular action in the present state. This requires knowledge of the **interventional distribution**, denoted using Pearl's 'do'-calculus as $P(s_{t+1}|s_t, \text{do}(a_t))$ (Pearl, 2009). The fundamental flaw of standard world models lies in the implicit and incorrect assumption that these two distributions are equivalent. In any

Figure 1: The Causal Disentanglement World Model (CDWM) separates dynamics into Environment and Intervention pathways, enabling robust planning and a causally-informed intrinsic reward (Agency Bonus).

environment with autonomous dynamics that might influence the agent's actions, confounding occurs, leading to a stark inequality:

$$\underbrace{P(s_{t+1}|s_t, a_t)}_{\text{Observational: What the model learns}} \quad \neq \quad \underbrace{P(s_{t+1}|s_t, \text{do}(a_t))}_{\text{Interventional: What the planner needs}} \quad . \quad (2)$$

A model that has learned mere correlation is doomed to fail under the **interventional distribution shift** that occurs when the agent's policy evolves. It cannot distinguish between the consequences of its actions and background events that just happened to co-occur, leading to catastrophic planning failures.

---

**Algorithm 1** Overall CDWM Training and Interaction Loop

---

1: **Initialize** models $h_\theta$, $g_\phi = (f_{\text{env}}, f_{\text{agent}})$, $f_\psi$, and replay buffer $\mathcal{D}$.
2: **for** episode = 1 to N **do**
3:     Receive initial observation $o_0$.
4:     **for** step $t = 0$ to $T - 1$ **do**
5:       $s_t \leftarrow h_\theta(o_{\leq t}, a_{<t})$
6:       Plan action $a_t$ using MCTS, which queries the model for $P(\cdot|s_t, \text{do}(a_t))$ (Alg. 2).
7:       Execute $a_t$, receive $(o_{t+1}, R_t, \text{done})$.
8:       Store observational transition $(o_t, a_t, R_t, o_{t+1})$ in $\mathcal{D}$.
9:       **if** $t \pmod{\text{train\_interval}} == 0$ and $|\mathcal{D}|$ is ready **then**
10:         Update parameters $\theta, \phi, \psi$ using a batch from $\mathcal{D}$ (see Sec. 3.4).
11:       **end if**
12:       **if** done **then break**
13:       **end if**
14:     **end for**
15: **end for**

---

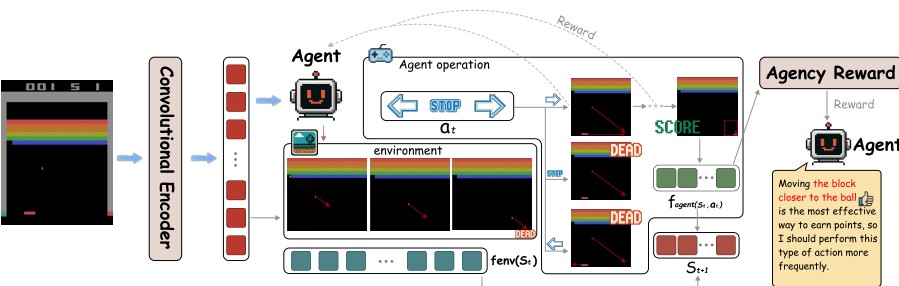

Figure 2: The architectural framework of the Causal Disentanglement World Model (CDWM). The model operationalizes a Structural Causal Model (SCM) to directly compute the outcome of interventions ($P(s'|s, \text{do}(a))$). The dual-path design enforces the separation of autonomous dynamics ($f_{\text{env}}$) from causal interventions ($f_{\text{agent}}$), enabling robust planning and principled, cause-driven exploration via the Agency Bonus.

This discrepancy gives rise to a fundamental technical challenge of **causal identifiability**. The interventional distribution $P(s_{t+1}|s_t, \text{do}(a_t))$ is not identifiable from the observational objective in Eq. equation 1 alone. Without further assumptions, there is no unique causal structure that corresponds to the observed data, preventing the model from learning the true effect of its actions. Our work confronts this identifiability problem head-on. We posit that the key to identification lies not in more complex statistical estimators, but in enforcing a structural constraint on the model itself. We propose the **Causal Disentanglement World Model (CDWM)**, illustrated in Figure 1, which makes the causal mechanisms identifiable by design. As detailed in Section 3, our architectural inductive bias serves as the necessary condition to disentangle confounded causal factors from purely observational data. The overall learning protocol is outlined in Algorithm 1.

## 3 METHODOLOGY: A STRUCTURAL SOLUTION TO CAUSAL CONFOUNDING

This section details our structural solution to the causal confounding problem. We begin by introducing the theoretical foundation of our approach: a **Structural Causal Model (SCM)** that provides a blueprint for disentanglement (Section 3.1). We then address the critical challenge of learning this SCM's components from confounded data, presenting our **dual-path architecture** as a targeted inductive bias that makes the causal mechanisms identifiable (Section 3.2). Building on this, we derive the **Agency Bonus**, a principled intrinsic reward enabled by our causal separation (Section 3.3), and finally, present the **joint optimization objective** for end-to-end training (Section 3.4).

### 3.1 A STRUCTURAL CAUSAL MODEL FOR LATENT DYNAMICS

Our core theoretical posit is a **Structural Causal Model (SCM)** (Pearl, 2009) for the latent dynamics. This SCM provides the theoretical recipe for computing the interventional distribution required for planning, thus directly addressing the challenge from Section 2.

**Assumption 1** (*Additive Causal Generative Process*). *The latent state $s_{t+1}$ is generated by a process where the total change in state is a linear superposition of an autonomous environmental dynamic and a direct agent-induced intervention. Formally, the structural assignment for $s_{t+1}$ is given by:*

$$s_{t+1} := s_t + \underbrace{f_{\text{env}}(s_t)}_{\text{Autonomous Dynamics}} + \underbrace{f_{\text{agent}}(s_t, a_t)}_{\text{Interventional Effect}} + \epsilon, \quad \text{where} \quad \epsilon \sim \mathcal{N}(0, \sigma^2 I). \tag{3}$$

**Theoretical Value.** This formulation is the cornerstone of our methodology because it analytically defines the interventional distribution. The term $f_{\text{agent}}(s_t, a_t)$ represents the **Total Causal Effect (TCE)** of action $a_t$. By learning the functions $f_{\text{env}}$ and $f_{\text{agent}}$, we are no longer fitting a black-box correlational model; we are explicitly learning the distinct causal mechanisms. This allows a planner to query the model for any counterfactual outcome, such as "what would have happened if I had done nothing?" ($s_t + f_{\text{env}}(s_t)$), providing the foundation for robust, causal-aware planning.

### 3.2 ARCHITECTURAL INSTANTIATION AND IDENTIFIABILITY

**Implementation Challenge.** While the SCM in Eq. equation 3 provides a theoretical blueprint, it introduces a formidable practical challenge: **identifiability**. From purely observational data, the functions $f_{\text{env}}$ and $f_{\text{agent}}$ are not uniquely identifiable. An optimizer minimizing the prediction error for $s_{t+1}$ has no principled way to correctly partition the observed state change $\Delta s_{t+1}$ between the two pathways, as countless combinations of the two functions could yield the same result.

**Our Solution.** To resolve this, we enforce identifiability through a hard **architectural constraint**. We design the dynamics model $g_\phi$ with a dual-path architecture that instantiates the two causal mechanisms with appropriate function classes. Specifically, the **Environment Pathway** $f_{\text{env}}$ is modeled using a Recurrent Neural Network (e.g., GRU), which is adept at capturing the temporal, self-evolving dynamics inherent to the environment's autonomous progression. In contrast, the **Intervention Pathway** $f_{\text{agent}}$ is modeled as a Multi-Layer Perceptron (MLP), which excels at approximating the complex but typically instantaneous mapping from a state-action pair $(s_t, a_t)$ to its direct consequence.

Crucially, we impose an **information bottleneck**: the action $a_t$ is architecturally forbidden as an input to the Environment Pathway (see Figure 2). This enforces the conditional independence $f_{\text{env}}(s_t) \perp a_t | s_t$ by design. This constraint is the instrument that breaks the non-causal "backdoor path" from action $a_t$ to the environmental component of the state transition, forcing the optimizer to attribute any action-independent dynamics to $f_{\text{env}}$ and isolating the pure causal effect of the action in $f_{\text{agent}}$. This architectural inductive bias is precisely what makes our SCM identifiable from observational data(see Appendix F for the identifiability theorem and proof)

### 3.3 QUANTIFYING CAUSAL INFLUENCE FOR INTRINSIC MOTIVATION

A direct and powerful consequence of achieving this architectural disentanglement is the ability to robustly quantify the agent's causal influence on the world. This provides a principled foundation for an intrinsic exploration bonus, which we term the **Agency Bonus**. Unlike traditional curiosity signals that are often confounded by environmental stochasticity, our bonus is derived directly from the isolated interventional component of our model. We define it as a normalized measure of the Total Causal Effect's magnitude:

$$r_t^{\text{i}} := \sigma \left( \frac{\|f_{\text{agent}}(s_t, a_t)\|_2 - \mu_\phi(s_t)}{\sigma_\phi(s_t) + \delta} \right), \tag{4}$$

where $\sigma(\cdot)$ is the sigmoid function, and $\mu_\phi(s_t)$ and $\sigma_\phi(s_t)$ are the running mean and standard deviation of the intervention magnitudes observed in states similar to $s_t$. This normalization ensures the bonus is a well-scaled signal that encourages the agent to explore actions leading to novel causal outcomes, rather than simply large or noisy state changes. The MCTS planner integrates this bonus as detailed in Algorithm 2.

---

**Algorithm 2** MCTS Planning with a Causal World Model

---

1: **procedure** MCTS-PLANNER($s_{\text{root}}, g_\phi, f_\psi$)
2: Initialize MCTS tree $\mathcal{T}$ with root node $s_{\text{root}}$.
3: **for** simulation = 1 to M **do**
4:    $s \leftarrow s_{\text{root}}$; path $\leftarrow [s]$
5:    **while** $s$ is an internal node in $\mathcal{T}$ **do**
6:      Select action $a$ to maximize PUCT score: $Q(s,a) + U(s,a)$.
7:      $(\hat{s}', \hat{r}) \leftarrow g_\phi(s,a)$    // Query the causal model
8:      $s \leftarrow \hat{s}'$; Append $(a, s)$ to path.
9:    **end while**
10:   $(\boldsymbol{p}, v) \leftarrow f_\psi(s)$
11:   Expand tree from node $s$ using policy priors $\boldsymbol{p}$.
12:   *// Compute causally-informed augmented value for backup*
13:   $r^{\text{i}} \leftarrow \text{ComputeAgencyBonus}(s, a, f_{\text{agent}})$ (using Eq. 4)
14:   $v_{\text{augmented}} \leftarrow \hat{r} + \lambda_{\text{agency}} \cdot r^{\text{i}} + \gamma v$
15:   Backup $v_{\text{augmented}}$ along the traversed path.
16: **end for**
17: **return** action $a_0$ from root with highest visit count $N(s_{\text{root}}, a_0)$.
18: **end procedure**

---

### 3.4 END-TO-END LEARNING VIA JOINT OPTIMIZATION

The preceding sections defined the causal structure and its architectural instantiation. We now specify the **joint optimization objective** that allows these components to be learned end-to-end. The entire model, $\Theta = \{\theta, \phi, \psi\}$, is optimized on trajectories from the replay buffer $\mathcal{D}$:

$$\mathcal{L}(\Theta) = \mathbb{E}_{\tau \sim \mathcal{D}} \left[ \sum_{k=0}^{K-1} \left( \mathcal{L}_{\text{policy}}^{(k)} + \lambda_v \mathcal{L}_{\text{value}}^{(k)} + \lambda_r \mathcal{L}_{\text{reward}}^{(k)} + \lambda_d \mathcal{L}_{\text{dynamics}}^{(k)} \right) \right]. \tag{5}$$

Crucially, the **Dynamics Loss** $\mathcal{L}_{\text{dynamics}}$ provides the supervision. For each observed transition $(s_t, a_t, s_{t+1})$, the loss minimizes the discrepancy between the ground-truth outcome $s_{t+1}$ and the prediction from our SCM, $s_t + f_{\text{env}}(s_t) + f_{\text{agent}}(s_t, a_t)$. This forces the two pathways, under the architectural constraint from Section 3.2, to learn their respective disentangled roles from the confounded, aggregated signal.

## 4 EXPERIMENT

### 4.1 BENCHMARK AND BASELINES

We conduct our primary evaluation on the **Atari 100k** benchmark (Ye et al., 2021a), a standard for assessing sample-efficient reinforcement learning. This suite comprises 26 classic Atari games that span a wide range of challenges, including reaction-based control, strategic planning, and goal-directed manipulation. Each agent is constrained to 100,000 environment interactions (equivalent to approximately 400,000 frames), a setup designed to rigorously test sample efficiency and policy generalization under a tight data budget.

Our primary evaluation metric is the **human-normalized score**, calculated as score = (agent − random)/(human − random). This metric contextualizes an agent's performance relative to both a random policy and an expert human player, providing a standardized measure of proficiency.

We benchmark CDWM against a comprehensive set of state-of-the-art model-based and model-free methods. These include early video-prediction models like **SimPLe** (Łukasz Kaiser et al., 2020); methods leveraging advanced representation learning techniques such as **SPR** (Schwarzer et al., 2021) and Transformer-based world models like **TWM** (Robine et al., 2023) and **IRIS** (Micheli et al., 2023). We also compare against the highly influential **DreamerV3** (Hafner et al., 2024) and its efficient Mamba-based variant **DramaXS** (Wang et al., 2025), as well as the Transformer-based **STORM** (Zhang et al., 2023). Finally, we include the current leading methods on this benchmark,

**BBF** (Schwarzer et al., 2023) and **EZ-V2** (Wang et al., 2024), which employ bootstrapped policies and Gumbel-based planning, respectively. Further details on these baselines are provided in Appendix B.

Table 1: Normalized performance comparison across 26 Atari100k (Ye et al., 2021a) tasks. The table reports game scores for each method using human-normalized metrics. **Bold** values indicate the best-performing method for each game, while underlined values denote the second-best. CDWM establishes a new state-of-the-art performance, achieving the highest Normalized Mean (2.81%) and Median (1.24%) scores, demonstrating superior sample efficiency and planning quality.

| Game | Random | Human | SimPLe | SPR | TWM | IRIS | STORM | DreamerV3 | DramaXS | BBF | EZ-V2 | CDWM (Ours) |
|---|---|---|---|---|---|---|---|---|---|---|---|---|
| Alien | 228 | 7128 | 617 | 823 | 675 | 420 | 984 | 959 | 820 | 1173 | **1558** | 1285 |
| Amidar | 6 | 1720 | 78 | 176 | 122 | 143 | 205 | 139 | 131 | 245 | 185 | **284** |
| Assault | 222 | 742 | 527 | 570 | 683 | 1524 | 801 | 706 | 539 | **2091** | 1758 | 1678 |
| Asterix | 210 | 8503 | 1128 | 969 | 1117 | 854 | 1028 | 932 | 1632 | 3946 | **61,810** | 23,765 |
| Bank Heist | 14 | 753 | 34 | 375 | 467 | 53 | 641 | 649 | 137 | 733 | 1317 | **1962** |
| BattleZone | 2360 | 37,188 | 5032 | 15642 | 5068 | 13,074 | 13540 | 12,250 | 10,860 | **24,460** | 14433 | 18456 |
| Boxing | 0 | 12 | 8 | 36 | 78 | 70 | 80 | 78 | 78 | 86 | 75 | **88** |
| Breakout | 2 | 30 | 16 | 18 | 20 | 84 | 16 | 31 | 7 | 371 | 400 | **462** |
| ChopperCommand | 811 | 7388 | 1078 | 965 | 1697 | 1565 | 1888 | 420 | 1642 | **7549** | 1197 | 2364 |
| CrazyClimber | 10,780 | 35,829 | 62,584 | 39,642 | 71,820 | 59,324 | 66,776 | **97,190** | 83,931 | 58,432 | 112,363 | 72,453 |
| DemonAttack | 152 | 1971 | 208 | 532 | 350 | 2034 | 165 | 303 | 201 | 13,341 | **22774** | 16742 |
| Freeway | 0 | 30 | 18 | 23 | 24 | 31 | 34 | 0 | 15 | 26 | 0 | **35** |
| Frostbite | 65 | 4335 | 242 | 1623 | 1476 | 259 | 1316 | 909 | 785 | 2385 | 1136 | **2554** |
| Gopher | 258 | 2412 | 645 | 682 | 1675 | 2236 | **8240** | 3730 | 2757 | 1331 | 3869 | 2765 |
| Hero | 1027 | 30,826 | 2657 | 6274 | 7254 | 7037 | 11,044 | 11,161 | 7946 | 7819 | 9705 | **14,896** |
| Jamesbond | 29 | 303 | 112 | 366 | 362 | 463 | 509 | 445 | 372 | **1130** | 468 | 579 |
| Kangaroo | 52 | 3035 | 238 | 3423 | 1240 | 838 | 4208 | 4098 | 1384 | **6615** | 1887 | 3972 |
| Krull | 1598 | 2666 | 2805 | 3684 | 6349 | 6616 | 8413 | 7782 | 9693 | 8223 | 9080 | **10642** |
| KungFuMaster | 258 | 22,736 | 16,541 | 13,762 | 24,555 | 21,760 | 26,183 | 21,420 | 23,920 | 18,992 | **28,883** | 25,766 |
| MsPacman | 307 | 6952 | 1480 | 1316 | 1588 | 999 | 2673 | 1327 | 2270 | 2008 | 2251 | **2860** |
| Pong | -21 | 15 | 13 | -5 | 19 | 15 | 11 | 18 | 15 | 17 | 21 | 21 |
| PrivateEye | 25 | 69,571 | 47 | 108 | 87 | 100 | 7781 | 882 | 90 | 41 | 100 | **10,296** |
| Qbert | 164 | 13,455 | 1289 | 735 | 3331 | 746 | 4522 | 3405 | 796 | 4447 | **16,058** | 13260 |
| RoadRunner | 12 | 7845 | 5641 | 13,622 | 9109 | 9615 | 17,564 | 15,565 | 14,020 | **33,427** | 27,517 | 18,692 |
| Seaquest | 68 | 42,055 | 683 | 558 | 774 | 661 | 525 | 618 | 497 | 1233 | 1974 | **2464** |
| UpNDown | 533 | 11,693 | 3350 | **23,162** | 15,982 | 3546 | 7985 | 7667 | 7387 | 12,102 | 15,224 | 18,765 |
| Normalised Mean (%) | 0 | 100 | 0.37 | 0.68 | 0.96 | 1.04 | 1.27 | 1.12 | 1.05 | 2.26 | 2.69 | **2.81** |
| Normalised Median (%) | 0 | 100 | 0.14 | 0.41 | 0.51 | 0.29 | 0.58 | 0.49 | 0.27 | 0.92 | 1.23 | **1.24** |

## 4.2 RESULTS AND ANALYSES ON ATARI100K

As shown in Table 1, our Causal Disentanglement World Model (**CDWM**) sets a new state of the art on the Atari100k benchmark. It achieves the highest aggregate scores across all methods in both **Normalized Mean (2.81%)** and **Normalized Median (1.24%)** performance, surpassing strong recent baselines like EZ-V2 (2.69%/1.23%) and BBF (2.26%/0.92%). These results strongly validate our central hypothesis: explicitly modeling the causal structure of world dynamics is a critical and highly effective strategy for enhancing sample efficiency and planning performance.

A deeper analysis reveals that CDWM's strengths are particularly pronounced in environments where disentangling causality is key. In games with complex, autonomous dynamics such as *Seaquest*, *Krull*, and *Frostbite*, CDWM achieves top-tier performance. By using its dual-path architecture to separate the environment's autonomous evolution from agent-induced changes, our model avoids causal confounding. This leads to more accurate long-horizon predictions and, consequently, more robust planning, as the agent can correctly attribute outcomes to their true causes.

Furthermore, CDWM demonstrates exceptional performance in control-intensive tasks where precise action consequences are paramount. For instance, in games like *Boxing*, *Breakout*, *Bank Heist*, and *MsPacman*, our model secures the top score. This success stems from the **Intervention Pathway**, which learns an isolated model of the agent's causal impact. The planner can thus rely on a high-fidelity understanding of what its actions will accomplish, enabling more effective and deliberate strategies. Notably, CDWM also excels in environments with extremely sparse and challenging reward landscapes. In *PrivateEye*, a notoriously difficult exploration task, our model achieves a score of 10,296, significantly outperforming all other methods. This highlights the practical benefit of our **Agency Bonus**, which provides a principled intrinsic motivation signal. By encouraging the agent to explore actions that have a tangible causal effect on the world, CDWM can systematically discover meaningful behaviors even in the prolonged absence of extrinsic rewards, a critical capability that monolithic world models lack. While not the top performer in every single game, CDWM exhibits highly competitive results across the entire suite, underscoring the general applicability and robustness of our causal disentanglement approach.

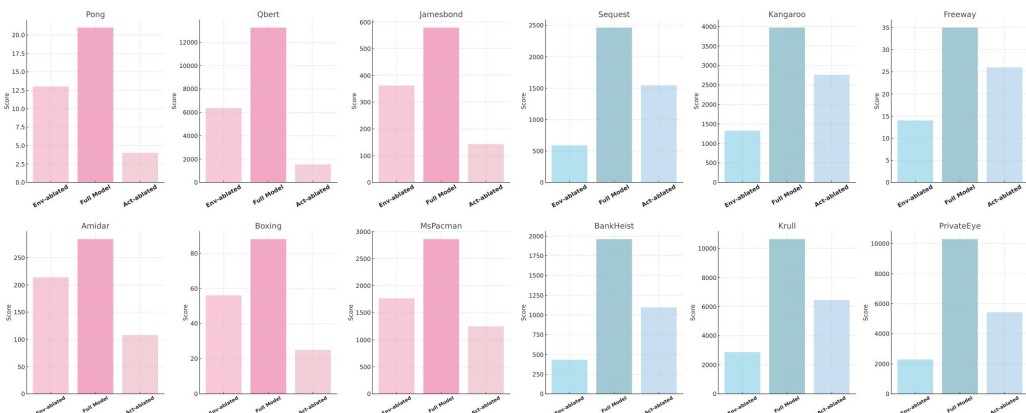

Figure 3: Ablation results of CDWM across control-dominant (red) and environment-dominant (blue) Atari tasks. Each plot compares the full model with variants removing either the environment pathway (Env-ablated) or the action pathway (Act-ablated). Control-dominant tasks show higher sensitivity to the agent pathway, while environment-dominant tasks rely more on environment modeling. Please zoom in for better viewing.

### 4.3 EVOLUTIONS OF INTERNAL REPRESENTATIONS

To assess whether CDWM can achieve *causal disentanglement* between the environment pathway (Env Pathway) and the action pathway (Act Pathway) in the representation space during training, we conduct a visualization analysis of the model's latent representations extracted at early, middle, and late training stages. Specifically, for each state-action pair, we obtain the corresponding latent vectors produced by the environment pathway $f_{\text{env}}(s_t)$ and the action pathway $f_{\text{agent}}(s_t, a_t)$, and project them onto a two-dimensional plane using t-SNE to examine the structural evolution of the two types of representations throughout training.

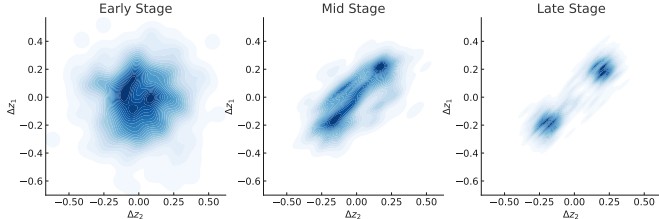

Figure 4: Evolution of disentangled representations in CDWM. The figure visualizes 2D projections of latent vectors from the environment pathway ($f_{\text{env}}$) and action pathway ($f_{\text{agent}}$) across three training stages. As training progresses, the initially entangled representations become increasingly separated, reflecting successful causal disentanglement. Please zoom in for better viewing.

As illustrated in Figure 4, during the **Early Stage** of training, the representations produced by the environment pathway and the action pathway are highly entangled, exhibiting no apparent structural distinction. This suggests that the model has not yet learned to differentiate the causal contributions of environmental context and agent actions to state transitions. In the **Mid Stage**, the two pathways begin to diverge along distinct directions, forming partially separated clusters. Although the boundary regions still contain a significant number of mixed points, this stage reveals an emerging trend toward semantic separation between the two sub-pathways. By the **Late Stage**, the two types of representations have evolved into well-separated distributions, each converging to a stable center in latent space. This reflects that the model has effectively learned to disentangle the subspaces corresponding to environmental dynamics and action effects. Nevertheless, a small degree of overlap remains in the intermediate regions, indicating that perfect orthogonal separation may not always be achievable in practice, due to potential nonlinear interactions between environment states and action outcomes. Overall, this experiment provides a clear structural perspective on how CDWM

progressively achieves representation disentanglement of different causal sources through pathway separation and causal modeling.

## 4.4 DISENTANGLING PATHWAY ROLES ACROSS TASK VARIANTS

To investigate whether our CDWM is capable of dynamically adjusting the relative contributions of the environment and action pathways based on task characteristics—thereby achieving better performance across different task types—we conduct a targeted analysis on two representative categories of Atari environments: **Control-dominant tasks**, where game dynamics are highly sensitive to agent actions: *Pong*, *Qbert*, *Jamesbond*, *Amidar*, *Boxing*, *MsPacman*; **Environment-dominant tasks**, where state transitions are primarily driven by external dynamics: *Seaquest*, *Kangaroo*, *Freeway*, *BankHeist*, *Krull*, *PrivateEye*.

For each task type, we evaluate a converged CDWM model under the following three configurations:

**Full Model**: both the environment and action pathways are active; **Env-ablated**: the environment pathway is deactivated, preserving only the action pathway; **Act-ablated**: the action pathway is deactivated, preserving only the environment pathway.

By measuring performance differences across these configurations while keeping all other conditions fixed, we assess the degree of dependence on each pathway and the model's ability to modulate their contributions adaptively across task types.

As shown in Figure 3, for control-dominant tasks, the model maintains high performance even after removing the environment pathway (i.e., **Env-ablated** configuration). For example, in the *Pong* task, the performance in the **Full Model** configuration is 21, while in the **Env-ablated** configuration it is 13, showing only a slight decrease in performance. However, when the action pathway is removed (i.e., **Act-ablated** configuration), performance drops significantly to 4. This indicates that the contribution of the environment pathway is relatively small in these tasks, and the action pathway plays a dominant role, proving the crucial importance of the action pathway in control-dominant tasks.

In contrast, for environment-dominant tasks, the contribution of the environment pathway is significantly higher than that of the action pathway. For instance, in the *Seaquest* task, the performance in the **Full Model** configuration is 2464, while in the **Env-ablated** configuration it drops to 589, showing a significant performance decline. On the other hand, when the action pathway is removed (i.e., **Act-ablated** configuration), the model's performance is 1546, with a smaller decrease. This demonstrates that the environment pathway is crucial for the performance in these tasks, and removing it significantly impacts the model's performance. These results show that CDWM adaptively balances environment and action pathways according to task characteristics

## 4.5 IMPACT OF AGENCY BONUS ON EXPLORATION EFFICIENCY

To evaluate the impact of agency-based intrinsic rewards on policy learning efficiency, we plot the learning curve that tracks the agent's average episodic return during the training. Specifically, we compare two configurations: (i) a full CDWM model with the proposed agency bonus; and (ii) an ablated variant with the same architecture and optimization setup but without the intrinsic reward, augmented with a constant offset to match the overall reward scale. All experiments are conducted in the sparse-reward Atari game *Venture*, which involves multiple sub-rooms with highly sparse reward signals, making it a suitable benchmark for assessing the agent's capacity for active state intervention. We focus on three key metrics. First, we record the average return within the initial 1,000 environment steps to assess whether the agent can trigger rewards in the early stage of training. Second, we measure the number of unique states visited during the same interval, using a latent-state-based hash to estimate exploration diversity. Finally, we track the trajectory of the agency bonus over the full 10,000-step training period to analyze its temporal dynamics and convergence behavior as a source of exploratory guidance.

Figure 5 illustrates the overall impact of the agency bonus on the agent's learning behavior and task performance. As shown in the top-left panel, agents equipped with the agency bonus achieve significantly higher average returns within the first 1,000 environment steps compared to the baseline without intrinsic rewards. The experimental group is able to trigger sparse environmental rewards

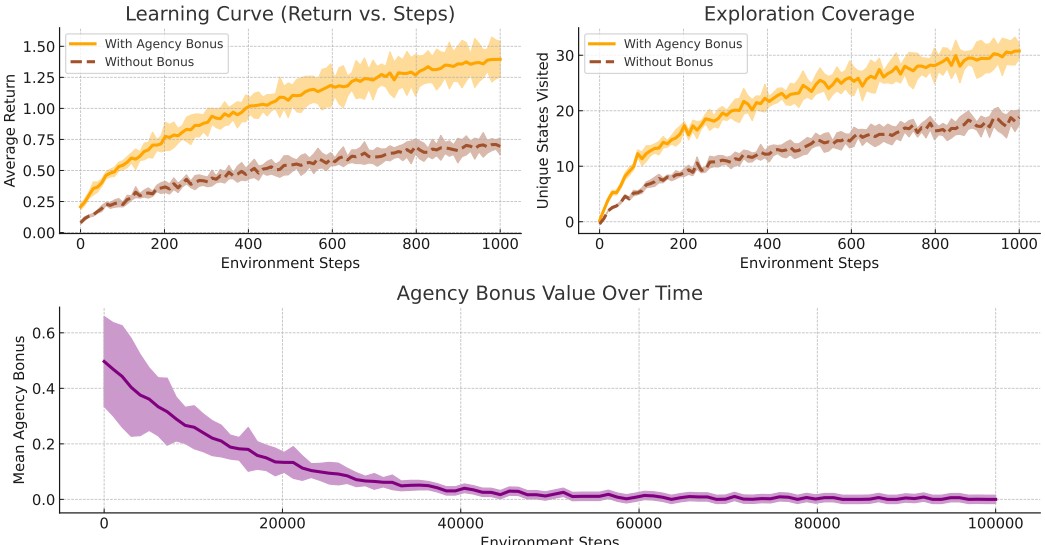

Figure 5: Impact of Agency Bonus on exploration and learning. Top: with the bonus, the agent gains faster returns (left) and explores more diverse states (right) in the first 1,000 steps. Bottom: the bonus peaks early then decays, giving strong initial exploration that fades as the policy converges

much earlier, with a learning curve that rises rapidly in the early stage, indicating that the agent quickly acquires task-relevant behavior. In contrast, the baseline remains nearly stagnant during the same period, with slow convergence, suggesting that it struggles to learn effective policies in the absence of exploratory incentives.

The top-right panel further presents the evolution of exploration coverage, measured as the number of unique states visited during training. The experimental group consistently maintains higher diversity in visited states within the first 1,000 steps, demonstrating that the agency bonus encourages the agent to explore alternative paths, sub-rooms, and novel regions, thereby avoiding local traps and repetitive behaviors. The baseline, by contrast, exhibits more concentrated and limited coverage, indicating a lack of proactive intervention and broader exploration.

The bottom panel shows the dynamics of the agency bonus throughout training. We observe that the intrinsic reward remains relatively high during the early stages, but gradually decays and stabilizes over time. This trend reflects a characteristic *early-shaping, late-annealing* behavior: the agency bonus serves as an initial exploration driver, helping the agent discover impactful actions, and then attenuates as the learned policy becomes more stable and goal-directed.

## 5 CONCLUSION

In this work, we introduced the Causal Disentanglement World Model (CDWM), a novel dual-path architecture that explicitly separates environment-driven dynamics from agent-induced interventions. Grounded in structural causal modeling, CDWM enables path-specific prediction and attribution, which we further harness via a cognitively inspired *Agency Bonus* to guide exploration toward actions with genuine and verifiable causal impact. Through extensive evaluation on the Atari100k benchmark, we demonstrated that our approach consistently outperforms state-of-the-art model-based methods in both sample efficiency and long-term planning accuracy, while also offering enhanced interpretability, robustness, and adaptability across control- and environment-dominant tasks. Key contributions of our work include: (i) a structural causal decomposition of state transitions into disentangled *environment* and *intervention* pathways, thereby promoting robust modeling, clearer attribution, and more faithful reasoning; (ii) the design and implementation of CDWM,which leverages this decomposition to stabilize multi-step prediction and planning under uncertainty; and (iii) the *Agency Bonus*, an intrinsic reward that quantifies an agent's causal influence, offers a reliable exploratory prior and accelerates learning in sparse-reward settings.

## ETHICS STATEMENT

This work adheres to the ICLR Code of Ethics. Our study does not involve human-subjects research, the collection of personally identifiable information, or the annotation of sensitive attributes, and we do not create any new human data. All experiments are conducted on the publicly available Atari 100k benchmark, strictly under its respective licenses and terms of use. As disclosed in the main paper, AI assistance was used responsibly in the preparation of this manuscript for literature summarization and language polishing to improve clarity.

## REPRODUCIBILITY STATEMENT

To ensure our results are fully reproducible, we have made the following provisions. **Code** The complete source code for the Causal Disentanglement World Model (CDWM), training scripts, and evaluation protocols will be made publicly available in a GitHub repository upon publication of this paper. **Data** All experiments are performed on the well-established and publicly accessible Atari 100k benchmark (Ye et al., 2021a), which ensures that the data is readily available to the research community. **Implementation Details** Appendix D provides a comprehensive account of all implementation details. This includes detailed network architectures for the encoder (Table 2), the dual-path dynamics model (Table 3), and prediction heads (Table 4). A complete list of hyper-parameters, which were held constant across all 26 games, is provided in Table 5. Furthermore, the computational costs required for training are detailed in Table 6 to allow for accurate resource planning.

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

## A    RELATED WORKS

**Model-Based Reinforcement Learning.** MBRL aims to enhance sample efficiency by learning a predictive world model, enabling planning with fewer real-world interactions (Luo et al., 2022; Plaat et al., 2020). Modern approaches have achieved remarkable success using latent state-space models. Methods like DreamerV3 (Hafner et al., 2024) and EfficientZero (Ye et al., 2021b) have set state-of-the-art benchmarks by leveraging sophisticated dynamics models and planning algorithms. Other lines of work have focused on improving the quality of learned representations, such as through self-supervision for temporal consistency (Schwarzer et al., 2021) or by learning invariant state abstractions (Micheli et al., 2023). A common thread in these advanced models, however, is the treatment of the dynamics function as a monolithic black-box. This forces the model to learn a purely correlational mapping, fundamentally limiting its ability to disentangle the underlying causal mechanisms of environmental change from the effects of agent interventions.

**Causality in Reinforcement Learning.** There is a growing recognition that integrating principles from causality can address fundamental challenges in RL, such as generalization and explainability (Zeng et al., 2025; Deng et al., 2023; Kaddour et al., 2022). Research in this area has explored causal inference for robust off-policy evaluation, learning disentangled representations by assuming independent causal mechanisms (Komanduri et al., 2024; 2022), and building explicit causal graphs for post-hoc interpretability (Yu et al., 2024; Madumal et al., 2020). While these methods pioneer the integration of causal reasoning, our work is distinct in that it embeds the causal structure—specifically, the separation of autonomous and intervened dynamics—directly into the forward prediction model used for planning. Rather than explaining decisions after the fact, our model uses its causal architecture to make better decisions from the outset.

**Intrinsic Motivation and Exploration.** To overcome challenges in sparse-reward environments, agents often rely on intrinsic motivation (Aubret et al., 2019; 2023). Classic approaches reward curiosity based on prediction error or state novelty (Burda et al., 2019; Pathak et al., 2017). A major drawback of these signals is that they are often confounded by environmental stochasticity; an agent may be rewarded for observing random, uncontrollable events rather than for its own meaningful actions. More recent work has begun to pursue causally-meaningful exploration signals (Nguyen et al., 2024; Yang et al., 2024). Our approach advances this line of inquiry in a novel way. In contrast to methods that learn a generic curiosity signal, our **Agency Bonus** is not learned but is derived directly from our model's architectural disentanglement. It provides a principled and explicit measure of the agent's causal influence, effectively isolating the reward signal from uncontrollable environmental factors.

## B    BASELINE MODEL DESCRIPTIONS

**SimPLe** (Łukasz Kaiser et al., 2020) learns a world model via video prediction, enabling policy optimization within a simulated environment to improve sample efficiency.

**SPR** (Schwarzer et al., 2021) learns temporally consistent latent representations through self-prediction, enhancing the data efficiency of model-free agents.

**TWM** (Robine et al., 2023) uses a recurrent latent-space model to learn long-horizon dynamics, generating virtual trajectories for policy optimization.

**IRIS** (Micheli et al., 2023) learns an invariant state abstraction by combining intrinsic rewards with a consistency-driven objective, focusing the agent on task-relevant features.

**STORM** (Zhang et al., 2023) improves model stability by eliminating the image decoder, instead learning a latent representation based solely on state-action prediction and reinforcement learning objectives.

**DreamerV3** (Hafner et al., 2024) is a leading RSSM-based agent that leverages discrete latent variables and an actor-critic framework to achieve high performance across a wide range of benchmarks.

**DramaXS** (Wang et al., 2025) is a lightweight world model that integrates the Mamba sequence model for highly efficient learning on benchmarks such as Atari100k.

**BBF** (Schwarzer et al., 2023) accelerates transfer learning by fine-tuning task-specific policies with a pretrained world model, using behavioral priors to guide adaptation without retraining the model itself.

**EZ-V2** (Wang et al., 2024) is a modular world model that uses high-resolution compression and low-latency policy networks to achieve strong performance and efficiency.

### B.1 FORMAL ANALYSIS OF MODEL BREAKDOWN UNDER NON-LINEAR DYNAMICS

To formalize the challenges outlined in Section E.1, we now provide a theoretical proof of how the CDWM's additive assumption leads to a breakdown in causal attribution and counterfactual reasoning when faced with non-linear dynamics, such as a 'mode switch'.

#### B.1.1 DEFINING A GROUND TRUTH WITH NON-LINEAR INTERACTIONS

Let's define a ground-truth Structural Causal Model (SCM) where the environment's dynamics depend on a latent mode variable, $m \in \mathcal{M}$. An agent's action can have a direct effect and can also change this mode. The true generative process is:

$$m_{t+1} = \mathcal{T}(m_t, a_t) \tag{6}$$

$$s_{t+1} = s_t + f_{\text{env}}^*(s_t, m_{t+1}) + f_{\text{agent}}^*(s_t, a_t) + \epsilon \tag{7}$$

where $\mathcal{T}$ is the mode transition function, and $f_{\text{env}}^*$ and $f_{\text{agent}}^*$ are the true, underlying causal mechanisms. A "mode-switching" action $a_{\text{switch}}$ is one for which $\mathcal{T}(m_t, a_{\text{switch}}) \neq m_t$. For all other actions $a'$, $\mathcal{T}(m_t, a') = m_t$.

#### B.1.2 PROOF OF CONFOUNDED CAUSAL ATTRIBUTION

Our model, CDWM, is constrained by its architecture to learn functions $f_{\text{env}}(s_t)$ and $f_{\text{agent}}(s_t, a_t)$ that minimize the predictive loss, typically the Mean Squared Error, with respect to the true outcome $s_{t+1}$:

$$\mathcal{L}_{\text{dyn}} = \mathbb{E} \left\| s_{t+1}^{\text{true}} - (s_t + f_{\text{env}}(s_t) + f_{\text{agent}}(s_t, a_t)) \right\|^2 \tag{8}$$

By substituting Eq. equation 7, the objective for the learned functions becomes minimizing:

$$\mathbb{E} \left\| \left( f_{\text{env}}^*(s_t, m_{t+1}) - f_{\text{env}}(s_t) \right) + \left( f_{\text{agent}}^*(s_t, a_t) - f_{\text{agent}}(s_t, a_t) \right) \right\|^2 \tag{9}$$

Due to the architectural constraint that $f_{\text{env}}$ is independent of $a_t$, it cannot adapt to the mode change $m_{t+1} = \mathcal{T}(m_t, a_t)$. The optimal $f_{\text{env}}(s_t)$ learned by the model will thus converge to an average of the true environmental dynamics over the distribution of modes it observes for state $s_t$:

$$f_{\text{env}}(s_t) \approx \mathbb{E}_{m_t \sim P(m|s_t)} \left[ f_{\text{env}}^*(s_t, m_t) \right] \tag{10}$$

Now, consider a mode-switching action, $a_t = a_{\text{switch}}$, which transitions the mode from $m_t$ to $m_t'$. To minimize the loss in Eq. equation 9, the entire burden of explaining the change in environmental dynamics must be absorbed by $f_{\text{agent}}$. The optimal learned function $f_{\text{agent}}$ for this action will be:

$$f_{\text{agent}}(s_t, a_{\text{switch}}) \approx f_{\text{agent}}^*(s_t, a_{\text{switch}}) + \left( f_{\text{env}}^*(s_t, m_t') - f_{\text{env}}(s_t) \right)$$

$$\approx f_{\text{agent}}^*(s_t, a_{\text{switch}}) + \underbrace{f_{\text{env}}^*(s_t, m_t') - \mathbb{E}_{m_t} \left[ f_{\text{env}}^*(s_t, m_t) \right]}_{\text{Causal Leakage Term}} \tag{11}$$

This formally proves that the learned interventional function $f_{\text{agent}}$ is no longer the true Total Causal Effect, $f_{\text{agent}}^*$. It is confounded by a **causal leakage term** that captures the difference between the new environmental dynamic and the model's average expectation. The principle of causal attribution is therefore violated.

#### B.1.3 PROOF OF FLAWED COUNTERFACTUAL REASONING

This misattribution directly leads to incorrect counterfactual predictions, which are essential for planning. Suppose the agent has just executed $a_{\text{switch}}$ at step $t - 1$, transitioning the environment to mode $m_t$. Now, at step $t$, the planner considers a counterfactual scenario, e.g., "what would happen if I were to perform a non-switching action $a_{\text{null}}$?"

The model's counterfactual prediction is generated using its learned components:

$$\hat{s}_{t+1}^{\text{cf}} = s_t + f_{\text{env}}(s_t) + f_{\text{agent}}(s_t, a_{\text{null}}) \tag{12}$$

The true counterfactual outcome, however, occurs in the current mode $m_t$:

$$s_{t+1}^{\text{true\_cf}} = s_t + f_{\text{env}}^*(s_t, m_t) + f_{\text{agent}}^*(s_t, a_{\text{null}}) \tag{13}$$

Assuming the model has learned the direct effect of the null action correctly ($f_{\text{agent}}(s_t, a_{\text{null}}) \approx f_{\text{agent}}^*(s_t, a_{\text{null}})$), the counterfactual error is:

$$\begin{aligned}
\text{Error}_{\text{cf}} &= \hat{s}_{t+1}^{\text{cf}} - s_{t+1}^{\text{true\_cf}} \\
&= f_{\text{env}}(s_t) - f_{\text{env}}^*(s_t, m_t) \\
&\approx \mathbb{E}_{m \sim P(m|s_t)} \left[ f_{\text{env}}^*(s_t, m) \right] - f_{\text{env}}^*(s_t, m_t) \tag{14}
\end{aligned}$$

This error term is non-zero unless the current environmental dynamic happens to be identical to the average dynamic. Since the agent has just forced a transition to a specific mode $m_t$, it is highly likely that this error is significant. The model's inability to condition its environmental predictions on the current mode leads to systematically biased counterfactuals and, consequently, flawed planning.

## C  COMPARATIVE ANALYSIS OF LEARNING DYNAMICS

While Table 4 establishes CDWM's superior final performance, it does not reveal how such advantages emerge throughout the training process. To offer a granular, step-by-step view into our model's behavior, we present a decomposition of the learning dynamics over time. This experiment moves beyond final scores to visualize how different design choices affect the efficiency and stability of learning.

Figure 6 presents a $3 \times 6$ grid in which rows denote key metrics and columns denote methods. All methods are evaluated on the sparse-reward Atari game `Venture` with five independent runs per method. Each subplot reports the mean trajectory (bold), a $\pm 1\sigma$ confidence band (shaded), and individual runs (transparent) over 100,000 interaction steps.

**Experimental Setup**  The six columns are: **(a–c): CDWM (Full)** – Our full model with dual-pathway + Agency Bonus. **(d–f): w/o Agency Bonus** – Removes the intrinsic reward to isolate its impact. **(g–i): DreamerV3** – A strong model-based RL baseline. **(j–l): EZ-V2** – A high-performing benchmark method. **(m–o): Env-ablated** – Removes the environment dynamics module. **(p–r): Act-ablated** – Removes the intervention pathway.

The three rows are: **Row 1 (Agency Bonus)** – Measures the magnitude of causal influence, only defined for CDWM variants. **Row 2 (Exploration Coverage)** – Counts unique states visited (via latent-state hashing). **Row 3 (Average Return)** – Standard task reward, measuring learning outcome.

**Key Observations**  **CDWM (Col 1)** demonstrates a well-structured learning trajectory. The Agency Bonus (Row 1, a) provides strong early guidance, then anneals as policy stabilizes. This drives broad state-space exploration (Row 2, b), which rapidly translates into performance gains (Row 3, c). **Without the bonus** (Col 2, d–f), the exploration is markedly weaker, with flatter coverage (e) and delayed returns (f), despite the same model architecture—highlighting that disentanglement alone is not sufficient in sparse-reward settings. **Baselines DreamerV3 and EZ-V2** (Cols 3–4, g–l) exhibit slower and less structured exploration. Their returns lag behind CDWM, especially in early training (i, l), revealing the benefit of causal guidance. **Architectural ablations** (Cols 5–6, m–r) result in collapsed learning: without either pathway, the model fails to explore (n, q) or accumulate meaningful rewards (o, r), confirming the necessity of both causal components. Figure 6 illustrates a causal learning pipeline: agency-aware dynamics enable targeted exploration, which leads to faster and more robust learning.

## D  ARCHITECTURE AND HYPERPARAMETER DETAILS

This appendix provides a comprehensive overview of the architectural and hyperparameter specifications used for all experiments in this work. To ensure full reproducibility, we detail the network

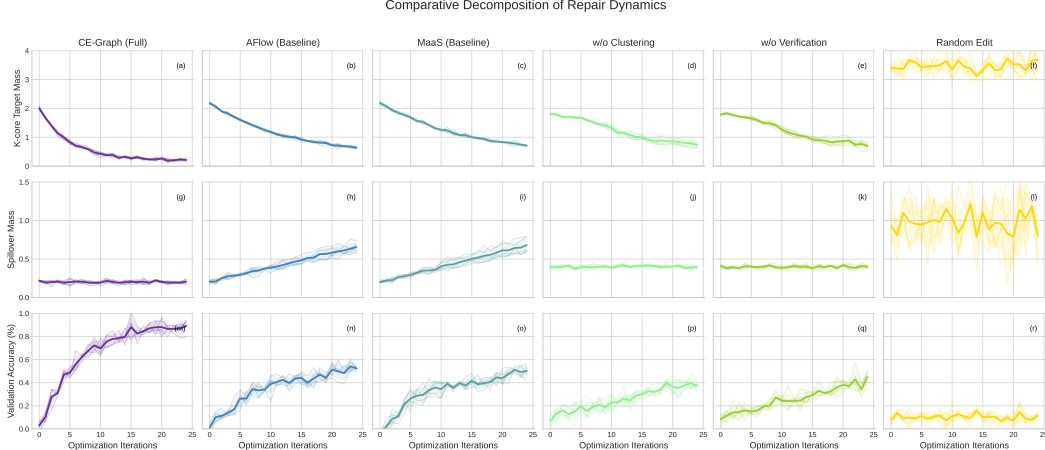

Figure 6: **Comparative Decomposition of Learning Dynamics on `Venture`.** Each subplot (a–r) compares learning curves of six methods (columns) across three metrics (rows). CDWM (Col 1) exhibits ideal behavior: an initially strong, decaying intrinsic bonus (Row 1) drives fast and diverse exploration (Row 2), leading to early and stable reward gains (Row 3). Removing the Agency Bonus (Col 2) severely degrades exploration and learning, while architectural ablations (Cols 5–6) cripple both model quality and planning effectiveness.

architectures for the core components of CDWM and list the hyperparameters, which were kept consistent across all 26 Atari100k environments for a fair and robust evaluation.

## D.1 MODEL ARCHITECTURES

The CDWM is composed of three main components: an image encoder that maps observations to a latent state, the core dual-path dynamics model that predicts future latent states, and several prediction heads for policy, value, and reward. Their respective architectures are detailed in Table 2 through Table 4.

Table 2: Image Encoder ($h_\theta$) Architecture. This network maps a stack of four $64 \times 64$ frames to a latent state vector $s_t$.

| Layer | Output Shape |
|---|---|
| Input Image Stack ($o_t$) | $12 \times 64 \times 64$ |
| Conv2d (k=4,s=2,p=1) + BatchNorm + ReLU | $32 \times 32 \times 32$ |
| Conv2d (k=4,s=2,p=1) + BatchNorm + ReLU | $64 \times 16 \times 16$ |
| Conv2d (k=4,s=2,p=1) + BatchNorm + ReLU | $128 \times 8 \times 8$ |
| Conv2d (k=4,s=2,p=1) + BatchNorm + ReLU | $256 \times 4 \times 4$ |
| Flatten | 4096 |
| Linear + LayerNorm | 512 |
| **Output:** Latent State ($s_t$) | $\mathbb{R}^{512}$ |

## D.2 HYPERPARAMETERS

A single, fixed set of hyperparameters was used to train the CDWM agent across all 26 Atari100k environments. The parameters for the general training loop, the world model, and the actor-critic planner are listed in Table 5.

Table 3: Dual-Path Dynamics Model ($g_\phi$) Architecture. Latent dimension $D = 512$, action dimension $A$ is game-specific.

| Submodule | Layer / Operation | Output Shape |
|---|---|---|
| **Environment Pathway ($f_{\text{env}}$)** | | |
| | Input State ($s_t$) | $D$ |
| | GRU Cell (Hidden Dim: $D$) | $D$ |
| | 2-Layer MLP (ReLU, Hidden Dim: $D$) | $D$ |
| **Intervention Pathway ($f_{\text{agent}}$)** | | |
| | Concatenate $[s_t, \text{onehot}(a_t)]$ | $D + A$ |
| | 2-Layer Gated MLP (GELU, Hidden Dim: $D$) | $D$ |
| **State Transition** | | |
| | $\Delta \hat{s}_{t+1} = f_{\text{env}}(s_t) + f_{\text{agent}}(s_t, a_t)$ | $D$ |
| | $\hat{s}_{t+1} = s_t + \Delta \hat{s}_{t+1}$ | $D$ |

Table 4: Prediction Head Architectures. All heads take the latent state $s_t$ as input.

| Head | Type | Input / Hidden / Output Dims |
|---|---|---|
| Policy Head ($\pi_\psi$) | 3-Layer MLP (ReLU) | $D$ / $D$ / $A$ |
| Value Head ($V_\psi$) | 3-Layer MLP (ReLU) | $D$ / $D$ / 1 |
| Reward Predictor ($R_\phi$) | 3-Layer MLP (ReLU) | $D$ / $D$ / 1 |

Table 5: CDWM Hyperparameters for the Atari100k benchmark.

| Group | Hyperparameter | Value |
|---|---|---|
| **General Training** | Replay Buffer Capacity | 100,000 |
| | Batch Size | 128 |
| | Unroll Length (Imagination Horizon) | 5 |
| | Optimizer | AdamW ($\beta_1 = 0.9, \beta_2 = 0.999$) |
| | Gradient Clipping Norm | 1.0 |
| **World Model** | Latent State Dimension ($D$) | 512 |
| | Dynamics Loss Weight ($\lambda_d$) | 2.0 |
| | Learning Rate | $3 \times 10^{-4}$ |
| **Actor-Critic** | Discount Factor ($\gamma$) | 0.997 |
| | MCTS Simulations per Step | 50 |
| | Value Loss Weight ($\lambda_v$) | 0.25 |
| | Reward Loss Weight ($\lambda_r$) | 1.0 |
| | Agency Bonus Weight ($\lambda_{\text{agency}}$) | 0.05 |
| | Learning Rate | $3 \times 10^{-5}$ |

### D.3 COMPUTATIONAL COST

To provide context on the model's efficiency, we report the approximate computational cost for training on the Atari100k benchmark. Following standard practice for fair comparison, all training times are estimated on a single NVIDIA V100 GPU equivalent, using the hardware conversion assumptions from DreamerV3 . An asterisk ($*$) denotes values extrapolated from hardware specifications reported in the original papers. A comparison with other state-of-the-art methods is provided in Table 6.

Table 6: Computational cost comparison on the Atari100k benchmark.

| Method | V100 Hours (Approx.) |
|---|---|
| SimPLe | 240$^*$ |
| SPR | 10 |
| TWM | 20$^*$ |
| IRIS | 168$^*$ |
| STORM | 9.3$^*$ |
| DreamerV3 | 12 |
| DramaXS | 7$^*$ |
| BBF | 13$^*$ |
| EZ-V2 | 16 |
| **CDWM (Ours)** | **11.7$^*$** |

## E LIMITATIONS AND FUTURE WORK

While our proposed CDWM framework sets a new state-of-the-art on the Atari100k benchmark and provides a robust method for causal disentanglement, we acknowledge certain limitations inherent in our core assumptions. This section discusses these limitations, focusing on the additive causal decomposition, and outlines promising directions for future research.

### E.1 ON THE LINEARITY OF THE CAUSAL DECOMPOSITION

The core of our methodology, as defined in **Assumption 1**, is the linear additive decomposition of state transitions presented in Eq. equation 3:

$$s_{t+1} := s_t + f_{\text{env}}(s_t) + f_{\text{agent}}(s_t, a_t) + \epsilon$$

This formulation, while powerful, is a first-order approximation of potentially more complex, non-linear environmental dynamics. We recognize this is a strong assumption, and its implications warrant a detailed discussion.

#### E.1.1 RATIONALE FOR THE ADDITIVE ASSUMPTION

Our choice of an additive structure was deliberate, motivated by several key factors that are crucial for enabling causal reasoning and stable learning: **Principled Causal Attribution:** The additive separation provides a clear and interpretable framework where the output of the Intervention Pathway, $f_{\text{agent}}(s_t, a_t)$, can be unambiguously interpreted as the isolated Total Causal Effect (TCE) of the agent's action. This clean separation is the bedrock upon which the Agency Bonus (Section 3.3) is built, as it allows us to measure causal influence directly. **Stable Inductive Bias:** Compared to modeling complex, unconstrained non-linear interactions, a linear superposition of functions imposes a stronger and more stable inductive bias on the learning process. This structure guides the optimization to find a meaningful separation of dynamics, mitigating issues like unstable gradients or degenerate solutions where one pathway might learn to ignore the other. **Effective First-Order Approximation:** For many dynamic systems, this formulation effectively captures the primary drivers of change—autonomous evolution and direct intervention. As our empirical results demonstrate, this approximation is sufficiently powerful for high-quality, long-horizon planning even in the complex visual environments of Atari games.

### E.1.2 Challenges in Scenarios with Non-Linear Interactions

Despite its strengths, the additive assumption faces challenges in environments where an agent's action induces a non-linear change in the world's dynamics, a phenomenon often referred to as a **'mode switch'**. For example, an action like flipping a light switch does not merely "add" light to the scene; it fundamentally alters the physical laws governing how the scene will evolve autonomously from that point forward.

**Misattribution of Effects:** The model's architecture forces it to account for the entire change in dynamics—including the shift from an old environmental function $f_{\text{env}}$ to a new one $f'_{\text{env}}$—entirely within the $f_{\text{agent}}$ term. This compels the Intervention Pathway to model not just the action's immediate effect, but also the long-term consequences of the altered environmental dynamics. This overburdens the MLP-based $f_{\text{agent}}$ and leads to an inaccurate, confounded model of both pathways.

**Failure of Counterfactual Planning:** A key benefit of our model is its ability to perform counterfactual queries, such as "what would have happened if I had done nothing?", which it computes as $\hat{s}_{t+1} = s_t + f_{\text{env}}(s_t)$. In a mode-switching scenario, this counterfactual is incorrect. After the switch is flipped, the true counterfactual should be based on the new dynamics $f'_{\text{env}}$. Because our model's $f_{\text{env}}$ is action-independent, it cannot adapt, leading to flawed predictions that would destabilize planning algorithms like MCTS.

**Potentially Misleading Agency Bonus:** The Agency Bonus is proportional to $\|f_{\text{agent}}(s_t, a_t)\|_2$. An action that causes a significant mode switch might correspond to a very small initial change in the state space (e.g., a single pixel changing color). This could result in a small output from $f_{\text{agent}}$ and thus a low agency bonus, causing the agent to ignore a critically important action for exploration.

### E.2 Future Work

The limitations of the additive model directly inspire several promising avenues for future research. Our primary goal is to extend the CDWM framework to capture non-linear, modulatory interactions between an agent's actions and the environment's dynamics. A promising direction involves designing architectures where the Intervention Pathway can influence the Environment Pathway's function. For instance, one could employ techniques like **HyperNetworks**, where $f_{\text{agent}}$ generates the weights for $f_{\text{env}}$, or **FiLM layers**, where $f_{\text{agent}}$ produces affine transformation parameters to modulate the features within $f_{\text{env}}$. Successfully modeling such interactions would represent a significant step toward more general and robust causal world models. Further research could also explore extending CDWM to partially observable 3D domains and multi-agent scenarios.

## F  Making Causal Effects Identifiable by Design

### F.1 Disentangling Environment and Agent Effects

The Causal Disentanglement World Model (CDWM), introduced in our work, addresses a critical challenge in Model-Based Reinforcement Learning (MBRL): the inability of traditional world models to distinguish between autonomous environmental dynamics and agent-induced interventions, leading to causal confounding. This issue, highlighted in Section 2, stems from the reliance on observational data to learn a black-box dynamics function, as seen in state-of-the-art methods like Dreamer (Hafner et al., 2020) and MuZero (Schrittwieser et al., 2020). These models excel at statistical prediction but falter under policy shifts due to their failure to capture causal mechanisms, a problem formalized as the mismatch between $P(s_{t+1} \mid s_t, a_t)$ and $P(s_{t+1} \mid s_t, \text{do}(a_t))$ (Pearl, 2009). Our dual-path architecture, detailed in Section 3.2, proposes a structural solution by enforcing a causal separation, but its effectiveness hinges on the identifiability of the learned components $f_{\text{env}}$ and $f_{\text{agent}}$ from observational data. This appendix provides a rigorous proof of that identifiability, building on the additive causal model (Assumption 1) and leveraging a null action to anchor the decomposition. The proof not only validates our architectural design but also advances the frontier by offering a principled approach to causal learning in MBRL, where prior work has largely avoided such structural constraints.

## F.2 NULL ACTION AS AN IDENTIFYING INSTRUMENT

(i) The data-generating process follows Assumption 1: $\Delta s := s_{t+1} - s_t = f^*_{\text{env}}(s_t) + f^*_{\text{agent}}(s_t, a_t) + \epsilon$, where $\mathbb{E}[\epsilon \mid s_t, a_t] = 0$ and $\epsilon$ is independent noise (e.g., Gaussian as in the model).

(ii) There exists a null action $a_0$ such that $f^*_{\text{agent}}(s_t, a_0) = 0$ for all $s_t$ (e.g., NOOP in Atari, which induces no change).

(iii) The behavior policy $\pi(a_t \mid s_t)$ has full support, meaning every action (including $a_0$) occurs with positive probability for each $s_t$ (as in $\epsilon$-greedy exploration).

(iv) The function classes for $f_{\text{env}}$ (e.g., GRU) and $f_{\text{agent}}$ (e.g., MLP) are universal approximators, trained to the global optimum under squared loss on observational data.

(v) The architecture enforces an information bottleneck: $f_{\text{env}}(s_t)$ has no access to $a_t$, ensuring $f_{\text{env}}(s_t) \perp a_t \mid s_t$ by design.

(vi) *Anchoring at the null action (training-side).* Either (a) we enforce the normalization $f_{\text{agent}}(s_t, a_0) \equiv 0$ as a hard architectural constraint, or (b) we include a vanishing Tikhonov penalty $\lambda \mathbb{E}[\|f_{\text{agent}}(s_t, a_0)\|^2]$ and take the limit $\lambda \to 0^+$. Both choices select the minimal-norm representative in the equivalence class of risk minimizers and coincide with the ground-truth anchor in (ii).

Then, the risk minimizer $(f_{\text{env}}, f_{\text{agent}})$ is unique and equals $(f^*_{\text{env}}, f^*_{\text{agent}})$ almost surely.

## F.3 PROOF OF THEOREM

We proceed step-by-step, grounding the argument in causal inference (Pearl, 2009) and statistical learning principles, with each step building toward the identifiability claim. This proof addresses a gap in the MBRL literature, where the identifiability of causal models from observational data has been underexplored, offering a theoretical foundation for our approach. **Step 1: Squared Loss Targets the Conditional Mean, Which is Interventional.**
Given Assumption 1 and the zero-mean noise condition $\mathbb{E}[\epsilon \mid s_t, a_t] = 0$, the expected state change is a fundamental quantity in our model, expressed as:

$$\Delta(s_t, a_t) := \mathbb{E}[\Delta s_t \mid s_t, a_t] = f^*_{\text{env}}(s_t) + f^*_{\text{agent}}(s_t, a_t). \tag{15}$$

The dynamics loss, a cornerstone of our training objective, is defined as:

$$L_{\text{dynamics}} = \mathbb{E}[\|\Delta s_t - f_{\text{env}}(s_t) - f_{\text{agent}}(s_t, a_t)\|^2], \tag{16}$$

and by the projection theorem in $L_2$ (Fisher consistency of the square loss), any global minimizer satisfies

$$f_{\text{env}}(s_t) + f_{\text{agent}}(s_t, a_t) = \Delta(s_t, a_t) \quad \text{a.s.} \tag{17}$$

Thus the model targets the conditional mean. From a causal perspective, $\Delta(s_t, a_t)$ aligns with the interventional quantity needed for planning. In the causal graph we have $s_t \to a_t$ (policy), $s_t \to s_{t+1}$ (via $f^*_{\text{env}}$), and $a_t \to s_{t+1}$ (via $f^*_{\text{agent}}$). Conditioning on $s_t$ closes the backdoor path $a_t \leftarrow s_t \to s_{t+1}$, yielding

$$P(s_{t+1} \mid s_t, \text{do}(a_t)) = P(s_{t+1} \mid s_t, a_t), \tag{18}$$

per the backdoor criterion. Hence minimizing equation 16 learns interventional effects rather than spurious correlations. **Step 2: The Null Action Anchors the Decomposition.**
By (ii),

$$\Delta(s_t, a_0) = f^*_{\text{env}}(s_t), \tag{19}$$

since $f^*_{\text{agent}}(s_t, a_0) = 0$. With full support (iii), the slice $a_0$ appears with positive probability for each $s_t$, so $\Delta(s_t, a_0)$ is empirically estimable. For any $a_t$,

$$\Delta(s_t, a_t) = \Delta(s_t, a_0) + f^*_{\text{agent}}(s_t, a_t), \tag{20}$$

and thus

$$f^*_{\text{agent}}(s_t, a_t) = \Delta(s_t, a_t) - \Delta(s_t, a_0). \tag{21}$$

This provides an anchor for the dual-path separation. **Step 3: The Architectural Bottleneck Enforces Unique Attribution at Risk Optimum.**

Fix $s_t$. Any global minimizer must satisfy

$$f_{\text{env}}(s_t) + f_{\text{agent}}(s_t, a_t) \;=\; \Delta(s_t, a_t) \quad \forall a_t, \tag{22}$$

with $f_{\text{env}}(s_t)$ independent of $a_t$ by (v). Evaluating equation 22 at $a_0$ yields

$$f_{\text{env}}(s_t) + f_{\text{agent}}(s_t, a_0) \;=\; \Delta(s_t, a_0). \tag{23}$$

At this point, statistical risk alone allows an additive shift $d(s_t)$ between paths (see Step 4). Assumption (vi) removes this degeneracy by *anchoring* the agent pathway at $a_0$: either the hard constraint $f_{\text{agent}}(s_t, a_0) \equiv 0$ or the vanishing penalty $\lambda \mathbb{E}\|f_{\text{agent}}(s_t, a_0)\|^2$ as $\lambda \to 0^+$ selects the minimal-norm representative. Under either choice, equation 23 forces

$$f_{\text{env}}(s_t) \;=\; \Delta(s_t, a_0) \;=\; f^*_{\text{env}}(s_t) \quad \text{a.s.} \tag{24}$$

and then

$$f_{\text{agent}}(s_t, a_t) \;=\; \Delta(s_t, a_t) - f_{\text{env}}(s_t) \;=\; f^*_{\text{agent}}(s_t, a_t). \tag{25}$$

**Step 4: Uniqueness of the Solution.**

Suppose another minimizer $(f'_{\text{env}}, f'_{\text{agent}})$ also attains the risk minimum, with

$$f'_{\text{env}}(s_t) = f^*_{\text{env}}(s_t) + d(s_t), \quad f'_{\text{agent}}(s_t, a_t) = f^*_{\text{agent}}(s_t, a_t) - d(s_t). \tag{26}$$

Without (vi), equation 26 preserves equation 22 for any $d(s_t)$. Under (via), the hard constraint $f_{\text{agent}}(s_t, a_0) \equiv 0$ implies $d(s_t) = 0$ by substituting $a_0$ in equation 26. Under (vib), the penalized risk adds $\lambda \mathbb{E}\|f'_{\text{agent}}(s_t, a_0)\|^2 = \lambda \mathbb{E}\|d(s_t)\|^2$, whose unique minimizer is $d(s_t) = 0$; taking $\lambda \to 0^+$ yields the same selection in the limit. Hence the minimizer is unique and equals $(f^*_{\text{env}}, f^*_{\text{agent}})$ almost surely. **Step 5: Convergence and Finite-Data Robustness.**

With infinite data, empirical conditionals converge:

$$\hat{\Delta}(s_t, a_t) \;\to\; \Delta(s_t, a_t) \quad \text{as } n \to \infty, \tag{27}$$

by the law of large numbers. Universal approximation (iv) ensures realizability of the target. For finite data, standard Rademacher generalization for square loss gives, with probability at least $1 - \delta$,

$$\|\hat{f}_{\text{env}} - f^*_{\text{env}}\|_{L_2(\mathcal{D}_s)} \;=\; O\!\left(\mathfrak{R}_n(\mathcal{F}_{\text{env}}) + \sqrt{\frac{\log(1/\delta)}{n_{a_0}}}\right), \tag{28}$$

where $\mathfrak{R}_n(\mathcal{F}_{\text{env}})$ is the Rademacher complexity of the environment class and $n_{a_0}$ is the effective sample size on the $a_0$-slice (scaling as $n \cdot \inf_s \pi(a_0 \,|\, s)$). An analogous bound holds for $f_{\text{agent}}$ uniformly over actions, with coverage factors $\inf_s \pi(a \,|\, s)$. This clarifies the practical role of exploration coverage in identifiability.

## STATEMENT ON THE USE OF AI ASSISTANCE

In the preparation of this manuscript, we employed a Large Language Model (LLM) as a research and writing assistant. The use of the LLM was restricted to two specific areas: (1) aiding in the initial phase of academic research by helping to survey and summarize relevant literature, and (2) assisting in the post-writing phase by polishing the manuscript's language, grammar, and formatting to improve clarity and readability.

