# OpenReview forum: "Causally Disentangled World Models: Guiding Exploration with an Agency Bonus"
_ICLR.cc/2026/Conference — ICLR 2026 Conference Withdrawn Submission_

### Official Review · Reviewer_EqGu · 2025-10-23

**Soundness:** 2
**Presentation:** 1
**Contribution:** 2
**Rating:** 2
**Confidence:** 4

**Summary:**

The paper proposes CDWM, a framework for a Causal Disentanglement World Model, which models state transitions in two ways: one is the traditional approach for learning, and the other is an interventional approach to identify causal effects. Experiments on the Atari 100K benchmark across more than 20 tasks show that CDWM achieves better performance compared to existing methods such as DreamerV3.

**Strengths:**

Clear idea: The work is well-motivated, approaching the problem from a causal inference perspective to improve the performance of world models, and proposes a clear two-pathway design to achieve this.

Experiments: The experiments cover a wide range of tasks, with comparisons to many recent relevant methods. Performance results presented across over 20 Atari tasks show that CDWM performs well.

**Weaknesses:**

1. In the abstract and introduction, the authors claim that the goal is to develop world models with high generalization and interpretability. However, CDWM is only evaluated on single-task performance, and neither the method design nor the experiments provide evidence that the proposed approach improves generalization or interpretability. The authors should clarify whether the stated research objectives are indeed achievable.

2. Concerns regarding core formula (Equation 3): In reinforcement learning, the state transition at time t+1 is determined by the current state and action according to the MDP framework. This formula does not appear to model transitions from the perspective of RL. Additionally, it is unclear what “Autonomous Dynamics” refers to, and how the “Interventional Effect” differs from the standard effect of taking an action—what is the role of the intervention here?

3. The paper mentions modeling via a Structural Causal Model (SCM), but it is not clear how the SCM is learned. How are the nodes and edges in the SCM designed? This section requires a more detailed explanation, and the authors could refer to relevant works on causal dynamics learning[1], IFactor[2].

4. Agentic reward design: The proposed agentic reward is not novel, as similar designs appear in works such as causal curiosity reward [3], causal efficiency [4] and ECL[5]. The authors should clarify the differences. Moreover, it is questionable whether simply introducing a single reward signal can truly improve policy generalization or interpretability. The method section should provide a clear and detailed explanation, from SCM modeling to the concrete Implementation of the algorithm.

5. Experiments: As mentioned above, additional experiments targeting generalization and interpretability are necessary to support the claimed research objectives.

6. typos: Line 182

The paper has an interesting idea, especially when reading the abstract and introduction, but the method description and experimental validation are not sufficiently thorough or convincing.

[1] Wang Z, Xiao X, Xu Z, et al. Causal Dynamics Learning for Task-Independent State Abstraction[C]//International Conference on Machine Learning. PMLR, 2022: 23151-23180.

[2] Liu Y, Huang B, Zhu Z, et al. Learning world models with identifiable factorization[J]. Advances in Neural Information Processing Systems, 2023, 36: 31831-31864.

[3] Sontakke S A, Mehrjou A, Itti L, et al. Causal curiosity: Rl agents discovering self-supervised experiments for causal representation learning[C]//International conference on machine learning. PMLR, 2021: 9848-9858.

[4] Seitzer M, Schölkopf B, Martius G. Causal influence detection for improving efficiency in reinforcement learning[J]. Advances in Neural Information Processing Systems, 2021, 34: 22905-22918.

[5] Cao H, Feng F, Fang M, et al. Towards empowerment gain through causal structure learning in model-based reinforcement learning[C]//The Thirteenth International Conference on Learning Representations. 2025.

**Questions:**

Please refer to Weaknesses part

---

### Official Review · Reviewer_Fzo3 · 2025-10-30

**Soundness:** 2
**Presentation:** 2
**Contribution:** 2
**Rating:** 2
**Confidence:** 4

**Summary:**

The paper studies how to separate self-dynamics from action-induced dynamics in model-based RL/world models, and then uses the action-induced part as a query to identify actions that cause large changes in the state. This signal is then used as an exploration bonus for policy learning. For policy learning, they run MCTS with the shaped reward. For the disentanglement, they use separate architectures to model the two dynamics. Empirically, the framework shows promising results on Atari compared with several world-model baselines.

Overall, the motivation is clear and reasonable. However, there are quite a few technical questions that are either not rigorously addressed or not fully explained in the current version. I will list them below. Because of these technical issues, I currently feel the paper is a bit below the ICLR bar, but I’m open to revising the rating after the authors clarify these points in the discussion/rebuttal

**Strengths:**

1. The motivation for separating self-dynamics from action-induced dynamics is clear and useful for world models, especially if the goal is to better understand action effects in the system.


2. The overall technical flow is clear. There might be some issues with the generic model form (Eq. 3), but in general the design looks reasonable.


3. The experimental evaluation on Atari looks solid.

**Weaknesses:**

Here I list the weaknesses and questions together, as many of them are closely related or overlapping.

W1: I think the most fundamental issue is the validity of Eq. (3), where you assume an additive decomposition of the dynamics. As far as I understand, if the goal is to get identifiability of the causal model, then even under more general post-nonlinear state dynamics [1] or with extra signals such as actions inducing significant state changes [2–3], identifiability is still possible in prior work. Here, however, the strict additive form seems stronger than what real dynamics (in RL) would satisfy, so there’s a gap between the assumption and what actually holds in practice.

W2: Echoing the above point, there seem to be several gaps. For instance, if $a_t$ affects $s_{1,t}$ and $s_{2,t-1}$ also influences $s_{1,t}$ you cannot simply model them additively, as this would violate the compounding effect of the physical dependencies both causes to the target. Since here you separate these effects into two independent networks, it is unclear how the compounded or interacting effects are actually captured or combined in the implementation.

W3: For the model, do you assume there is no temporal correlation between the noise factors? In many RL settings, the noise inherently carries information, e.g., environmental effects like wind changes are temporally correlated but may not be explicitly represented in the states. To properly capture the underlying causal process [4], I believe this aspect should also be taken into account.

W4: Here, regarding the “intervention” path, it seems more like an action-dynamics network rather than a true intervention. You are not actually intervening on specific state variables in the system — the action is sampled from the policy, not applied as a do-operator in the causal sense as suggested in Eq. (1), right?

W5: For the MCTS part, it’s a bit unclear what the causal model or function $g$ you mentioned (the one being queried) actually refers to. I assume it represents the entire dynamics model, but it would be good to clarify this explicitly.

W6: For the reward shaping term, it feels quite similar to curiosity-based or ensemble-error methods [5–6], since both aim to find actions that cause large state changes. Although the mathematical form here is different, the underlying idea seems related. In that case, what is the actual connection to the notion of causality in this work?

W7: In the experiments, it would also be more rigorous to add the same reward-shaping term to the baselines to check whether they can benefit from it as well.

W8: I checked the appendix, and it seems you are using a different architecture for the latent dynamics than Dreamer (I mean the RNN part, not the disentanglement). It would be more rigorous to align the architectures for a fair comparison. Also, since different world models now use quite different backbones, especially transformer-based ones, it would be helpful to include a version with such an architecture to show that your framework is generic, easy to integrate, and architecture-agnostic.

W9: What’s the necessity for using MCTS here? Why not align more closely with Dreamer by directly learning a policy and using the learned model to imagine additional data for training? It’s not clear what unique benefit MCTS brings in this setup compared to standard policy optimization with imagination-based rollouts.

W9: For ablation, better to consider more types of intrinsic motivations as reward shaping terms, such as [5-6].

[1] Uemura, Kento, et al. "A multivariate causal discovery based on post-nonlinear model." Conference on Causal Learning and Reasoning. PMLR, 2022.

[2] Von Kügelgen, Julius, et al. "Self-supervised learning with data augmentations provably isolates content from style." Advances in neural information processing systems 34 (2021): 16451-16467.

[3] Yao, Weiran, Guangyi Chen, and Kun Zhang. "Temporally disentangled representation learning." arXiv preprint arXiv:2210.13647 (2022).

[4] Swamy, Gokul, et al. "Causal imitation learning under temporally correlated noise." International Conference on Machine Learning. PMLR, 2022.

[5] Kauvar, Isaac, et al. "Curious Replay for Model-based Adaptation." International Conference on Machine Learning. PMLR, 2023.

[6] Sekar, Ramanan, et al. "Planning to explore via self-supervised world models." International conference on machine learning. PMLR, 2020.

**Questions:**

I listed most questions with weakness in the above section. Here are a few more:

1. For the reward shaping term, could this lead to training instability? Since the agent is constantly encouraged to take actions that cause the most significant state changes, this might conceptually push the policy toward unstable or chaotic behaviors.

2. Figure 4 is a nice visualization of the disentanglement factors, but it’s not a direct demonstration. Since you’re using reconstruction, why not show the reconstructed outputs for each factor separately?

3. Do you have any additional assumptions on the action space? Can this framework generalize to continuous actions as well, since the current experiments on Atari only involve discrete ones?

---

### Official Review · Reviewer_2Sh8 · 2025-10-31

**Soundness:** 3
**Presentation:** 3
**Contribution:** 2
**Rating:** 4
**Confidence:** 3

**Summary:**

This paper proposes the Causally Disentangled World Model (CDWM) for model-based RL. The key idea is to structurally separate latent dynamics into an Environment Pathway which models action-independent, autonomous dynamics, and an Intervention Pathway, which models the direct effect of actions. The transition equation is an additive model, and implemented with an architectural constraint that actions do not enter the environment pathway. The authors argue this “causal factorization” makes interventional predictions identifiable from observational data and improves planning. Building on the decomposition, the paper defines an Agency Bonus—an intrinsic reward proportional to the magnitude of the intervention pathway output (normalized per-state)—to encourage causally impactful exploration. The method is evaluated on Atari100k (26 games, 100k steps ≈ 400k frames), reporting state-of-the-art performance against strong model-based and model-free baselines, with ablations on pathway removal and on the Agency Bonus.

**Strengths:**

* The dual‑path constraint is a simple architectural prior that plausibly reduces action‑related confounding in practice and yields a neat handle for intrinsic motivation.

* Exploration in sparse‑reward games. The Agency Bonus is interesting and, if properly calibrated, more targeted than generic curiosity. The ablation results of agency bonus are particularly encouraging.

* The pathway‑drop ablations are informative, showing different sensitivity on “control‑dominant” vs “environment‑dominant” games gives credibility to the intended decomposition.

* The t‑SNE trajectories suggest the two pathways learn distinct codes over training, aligning with dual-path constraint for isolating environment part and control part in the data generation process.

**Weaknesses:**

* The theoretical part is built upon a fully observed MDPs, where the methodology and experiments seem to be conducted in a POMDP way. There would be unobserved confounders in the POMDP, which makes the identifiability results not hold. The connection and identifiability proof to the POMDP case need to be made for a complete theoretical guarantees.

* Disentangled world models have been studied extensively, and it is highly relevant to the environment and control dual-path learning in this work, e.g., [1][2][3]. The author should add a discussion and position this work considering them.

* The experiment section compares mostly world models and misses intrinsic reward baselines. The author may consider adding an intrinsic reward baseline, e.g., [4][5][6].

* Besides efficient exploration, this paper is motivated by poor generalization of reinforcement learning, but it seems no experiments and analysis related to the generalization of the proposed approach. Since the paper emphasizes robustness under policy shift and deconfounding, add stress tests where observational correlations break (e.g., swapped opponent dynamics, altered exogenous event rates) to show the benefit of the separation.

[1] Liu, Yuren, et al. "Learning world models with identifiable factorization." Advances in Neural Information Processing Systems 36 (2023): 31831-31864.

[2] Wang, Xinyue, and Biwei Huang. "Modeling Unseen Environments with Language-guided Composable Causal Components in Reinforcement Learning." The Thirteenth International Conference on Learning Representations.

[3] Wang, Tongzhou, et al. "Denoised mdps: Learning world models better than the world itself." arXiv preprint arXiv:2206.15477 (2022).

[4] Mendonca, Russell, et al. "Discovering and achieving goals via world models." Advances in Neural Information Processing Systems 34 (2021): 24379-24391.

[5] Ferrao, Jeremias, and Rafael Cunha. "World Model Agents with Change-Based Intrinsic Motivation." arXiv preprint arXiv:2503.21047 (2025).

[6] Bharadhwaj, Homanga, et al. "Information prioritization through empowerment in visual model-based RL." arXiv preprint arXiv:2204.08585 (2022).

**Questions:**

I have left my questions and suggestions in the weaknesses section, please refer to it.

---

### Official Review · Reviewer_Huyv · 2025-11-01

**Soundness:** 2
**Presentation:** 4
**Contribution:** 3
**Rating:** 4
**Confidence:** 5

**Summary:**

This paper addresses causal confounding in model-based reinforcement learning, where standard world models learn observational distributions $P(s_{t+1}|s_t,a_t)$ rather than the interventional distributions $P(s_{t+1}|s_t,\text{do}(a_t))$ needed for robust planning. The authors propose CDWM (Causal Disentanglement World Model), which decomposes state transitions via an additive structural causal model:
\[
s_{t+1} := s_t + f_{\text{env}}(s_t) + f_{\text{agent}}(s_t,a_t) + \epsilon.
\]
By architecturally forbidding actions from entering $f_{\text{env}}$, the model enforces $f_{\text{env}}(s_t) \perp a_t \mid s_t$, isolating autonomous dynamics from agent-induced interventions. The magnitude of $f_{\text{agent}}$, termed the Total Causal Effect (TCE), drives an ``Agency Bonus'' intrinsic reward to guide exploration toward causally impactful actions. On Atari-100k, CDWM achieves top aggregate performance (2.81\% normalized mean), with ablations demonstrating task-specific functional specialization of the two pathways.

**Strengths:**

This work has several notable strengths that merit recognition:

1. The problem is clearer formulated and well motivated. The paper articulates the distinction between observational and interventional distributions precisely and motivates why causal confounding undermines planning under policy shifts. The framing is pedagogically clear.

2. CDWM provides a simple and interpretable architectural constraint. The hard constraint $f_{\text{env}}(s_t) \perp a_t \mid s_t$ is elegant and minimal. It provides a concrete inductive bias that is easy to implement and reason about.

3. The authors provide strong task-specific ablation evidence as show in Figure 3 where pathway ablations are particularly compelling. When the agent pathway is removed, performance collapses on \emph{control-dominant} games (Pong, QBert, Boxing) where precise actions drive success. When the environment pathway is removed, \emph{environment-dominant} games (Seaquest, Kangaroo, Frostbite) with strong autonomous dynamics degrade sharply. If the decomposition were arbitrary, removing either branch would cause roughly uniform degradation. Instead, the failure modes align exactly with task structure, demonstrating that the model has learned to \emph{allocate explanatory power to the correct causal mechanism}. The two pathways are not redundant---each captures a distinct, functionally meaningful process. This is strong empirical evidence for causal disentanglement within the learned representation.

4. They provide emergent representation disentanglement as shown in Figure 4. Figure 4's t-SNE visualizations show that $f_{\text{env}}$ and $f_{\text{agent}}$ representations are entangled early in training but progressively separate into distinct clusters. This self-organization occurs \emph{without explicit disentanglement losses}, suggesting the architectural constraint alone provides sufficient inductive bias. The late-stage separation validates that the model is learning the intended factorization.

5. The authors provide a principled exploration with self-annealing behavior. The Agency Bonus (Figure 5, bottom panel) exhibits the signature of a causal-influence measure: it peaks early when the policy probes causally potent actions, then naturally decays as those effects become predictable. This self-annealing property distinguishes it from noise-driven curiosity signals and aligns with the theoretical motivation. The top panels show that this translates to faster learning and broader state coverage.

6. They demonstrate SOTA results, achieving top aggregate performance on Atari-100k (Table 1: 2.81\% mean, 1.24\% median vs.\ EZ-V2's 2.69\%/1.23\%) with computational cost comparable to DreamerV3 ($\sim$11.7 V100 hours, Table~6).

7. The Limitations section (Appendix E) explicitly discusses when the additive assumption fails (mode-switching, light switch examples). Fixed hyperparameters across all 26 games, detailed architectures (Tables~2--5), and a commitment to release code facilitate replication. This transparency is appreciated.

**Weaknesses:**

1. The additive assumption is fundamentally restrictive. The core equation $s_{t+1} = s_t + f_{\text{env}}(s_t) + f_{\text{agent}}(s_t,a_t) + \epsilon$ is acknowledged as a "first-order approximation" (Appendix E), but I believe this undersells the severity of the limitation. The authors prove (Appendix~B.1.3) that under mode-switching dynamics, where actions alter the environment's evolution, the learned $f_{\text{agent}}$ becomes confounded with environmental state changes, yielding systematic counterfactual bias. More importantly, no quantitative evidence is provided for when/where additivity holds. The ablations demonstrate the decomposition works within Atari's discrete action space, but many realistic dynamics involve multiplicative interactions (friction $\propto$ velocity, lighting changes, contact forces), threshold effects, or compositional structure that fundamentally violate linear superposition. The honest discussion of this limitation is appreciated, but it constrains the generalizability of the approach. Without evidence that the additive assumption is broadly valid, or an extension to handle non-additive interactions, the causal claims remain domain-specific.

2. They present observational training without interventional validation. The model is trained exclusively on transitions $(s_t, a_t, s_{t+1})$ sampled under behavior policies, standard supervised learning with architectural constraints. While Appendix B.1.3 provides theoretical analysis of counterfactual error under mode-switching, there are no empirical intervention experiments where actions are externally imposed and $f_{\text{agent}}$'s predictions verified against ground-truth $P(s'|s,\text{do}(a))$. Recent work in causal RL includes such validation, as seen in language-grounded causal world models[1], where they report 1-step and N-step intervention accuracy independently of task returns. Additionally, [2] provides generalization error bounds and validates the learned causal structure via conditional independence tests. While the ablations provide strong functional evidence, direct interventional metrics would more conclusively demonstrate that $f_{\text{agent}}$ captures $P(s'|s,\text{do}(a))$.

3. All results are confined to Atari-100k; a single domain with discrete actions and frame-based observations. The paper claims to address a general problem in world modeling (e.g., causal confounding) yet provides evidence only within Atari's narrow context. The additive assumption that works for pixel-based discrete actions may catastrophically fail in continuous control with contact dynamics, articulated robots, or fluid interactions. Recent work in causal world modeling [2] demonstrates generalization across robotics domains (Meta-World) with compositional transfer. CDWM provides no such evidence.  Without multi-domain validation, the contribution reads as ``a factorization trick that improves Atari scores.''

4. The paper's language, promises causal inference. What is delivered is an architectural factorization that demonstrably organizes representations but lacks interventional validation. This gap manifests in several ways: (1) Identifiability claims are architectural, not causal. The proof (Appendix F) requires strong domain-specific assumptions: (i) null action existence (NOOP), (ii) full support, (iii) universal approximation, (iv) global optimization. It establishes parameter identification under the assumed additive structure, but not that the structure itself is causally correct.

5. The Agency Bonus $r^i_t = \sigma\big((\|f_{\text{agent}}\|_2 - \mu)/\sigma\big)$ rewards large interventions. While the self-annealing behavior (Fig. 5) is encouraging, the derivation is heuristic rather than principled. A more formal connection to information-theoretic notions (empowerment, causal influence measures) would strengthen the theoretical foundation.

[1] Gkountouras, J., Lindemann, M., Lippe, P., Gavves, E., and Titov, I. Languageagents meet causality – bridging llms and causal world models. In Proceedings of the International Conference on Learning Representations (ICLR) 2025 (2025).

[2] Wang, X., and Huang, B. Modeling unseen environments with language-guided composable causal components in reinforcement learning. In Proceedings of the International Conference on Learning Representations (ICLR) 2025 (2025). Poster session.

**Questions:**

1. Can you provide empirical evidence demonstrating when the additive decomposition ($s_{t+1} = s_t + f_{\text{env}}(s_t) + f_{\text{agent}}(s_t, a_t) + \epsilon$) holds in practice?
2. Have you tested the model under explicit interventions to verify that ($f_{\text{agent}}) predicts (P(s' \mid s, do(a))$)?
3. Do you have results on another domain, such as DMC, MuJoCo, or Meta-World, to test robustness beyond Atari?
4. How can the degree of representation disentanglement between ($f_{\text{env}}) and (f_{\text{agent}}$) be quantified?
5. How sensitive is model performance to the value of ($\lambda_{\text{agency}}$) in the intrinsic-reward term?
6. Is there a theoretical connection between the Agency Bonus and information-theoretic influence measures like empowerment?
7. Would you consider reframing the claims as causally inspired architectural disentanglement rather than full causal inference?

**Details Of Ethics Concerns:**

No concerns.

---

### Note · Authors · 2025-11-12

**Comment:**

I have read and agree with the venue's withdrawal policy on behalf of myself and my co-authors.

**Withdrawal Confirmation:**

I have read and agree with the venue's withdrawal policy on behalf of myself and my co-authors.